# The Influence of Microwave Treatments on Bioactive Compounds and Antioxidant Capacity of *Mentha piperita* L.

**DOI:** 10.3390/ma15217789

**Published:** 2022-11-04

**Authors:** Livia Bandici, Alin Cristian Teusdea, Vasile Darie Soproni, Francisc Ioan Hathazi, Mircea Nicolae Arion, Carmen Otilia Molnar, Simona Ioana Vicas

**Affiliations:** 1Department of Electrical Engineering, University of Oradea, 410087 Oradea, Romania; 2Department of Environmental Engineering, University of Oradea, 410087 Oradea, Romania; 3Department of Food Engineering, University of Oradea, 410087 Oradea, Romania

**Keywords:** *Mentha piperita* L., microwave processing, polyphenols, flavonoids, DPPH, FRAP, chemometric multivariate analysis

## Abstract

Microwave extraction is becoming a popular option in many fields, especially for bioactive compounds from medicinal plants. This paper addresses the application of microwaves in the process of extracting bioactive compounds (phenols, flavonoids, chlorophyll) from peppermint with antioxidant capacity in order to highlight the influence of the microwave field on the quality of the final product in comparison with the control samples. The *Mentha piperita* L. is a rich source of phenols. The total phenol content after applying the MW treatments significant increased and varied between 25.000 ± 1.992 and 391.687 ± 20.537 mg GAE/100 g dw compared to the untreated sample (8.089 ± 2.745 mg GAE/100 g dw). The same trend was also recorded in the case of the flavonoid and pigment content in peppermint leaves following the application of microwave treatments. The obtained results were investigated using chemometric multivariate analysis. The main purpose of our research was to compare the possibilities of total or partial substitution of conventional extraction technologies with the microwave extraction technology, and also to highlight the existing differences in the amount of total phenols and flavonoids extracted from peppermint plants in different processing conditions. Through microwave processing, a significant increase in polyphenolic compounds is obtained.

## 1. Introduction

Following new research, medicinal plants have been brought back to actuality and considered valuable for the pharmaceutical industry, which has led to the fact that lately they are used more and more often in the prevention and cure of numerous diseases that affect the human body. Romania is one of the richest countries in medicinal and aromatic plants [1]. Due to the varied climatic conditions, as well as the multiple landforms and soil types, and other favourable conditions, more than 300 species of medicinal plants can be found in our country. 

Peppermint constituents are mostly studied in relation to their essential oils, which are used in a wide variety of industries. The global trade in essential oils has expanded, due to the large number of consumers and the widening spectrum of uses of these compounds [2,3]. In addition to these compounds, peppermint also contains polyphenols with important biological properties. The genus *Mentha* is rich in phenolic acids such as caffeic acid and its derivatives, chlorogenic and rosmarinic acid, the latter representing the majority of the total phenolic compounds. Besides these, salvianolic acids were also detected in peppermint [4]. Peppermint leaves are rich in flavonoids, from the class of flavones and flavanones. Luteolin and its derivatives, eriocitrin, naringenin-7-O-glucoside, and apigenin are the main flavonoids described in *Mentha* species [4,5,6]. In another study, the major flavonoids identified in peppermint were hesperidin, luteolin, and kaempferol [7]. Different pigments such as carotenoids [8] and chlorophylls (chlorophylls a and b) have also been identified in Mentha species [9,10] and among vitamins, ascorbic acid and α-tocopherols have been reported in peppermint leaves [4,8,10]. In a recent study, the authors evaluated thirteen *Mentha* species and found that the level of phenolic acids did not significantly differ, instead the levels of flavonoids varied within the species [7]. Ramawat (2009) [11] estimates that about 30% of the number of new substances commercialized between 1981 and 2006 are synthetic, only 70% being natural substances. 

At present, there is a major concern from researchers for the development of efficient, rapid, and environmentally friendly extraction methods that are capable of reducing extraction time, solvent consumption, and significantly reducing environmental pollution and energy costs [12]. Microwave-assisted extraction (MAE) is a technique in which microwaves are applied to accelerate the extraction of compounds, for example from peppermint leaves, thus achieving a higher extraction speed with reduced processing costs. The microwaves heat the cell walls, which allows the component compounds to flow into the organic solvent used [13]. Kohari et al., (2020) [14] applied solvent-free microwave extraction to *Mentha arvensis* L. in order to obtain essential oils and demonstrated that this method is more efficient than traditional hydrodistillation extraction. The fragrance components such as limonene and piperitone were found in a higher amount in microwave extraction compared with the traditional method [14,15]. A microwave-assisted extraction method has been optimized by using ethanol as a solvent to extract terpenoids from *Mentha rotundifolia* [6]. Carvone, a monoterpene that is present in peppermint, was the major compound extracted using microwaves [16]. 

*Mentha* species are characterized by great chemical diversity and were reported to contain a number of chemical compounds responsible for pharmacological properties. Among the beneficial effects of peppermint can be listed the antioxidant, antimicrobial, antiviral, and anti-carcinogenic effects [17] in addition to antiseptic, anti-inflammatory, anti-fermentative, analgesic, disinfectant, spasmolytic, sedative, decongestant, and expectorant properties. Thymol, a phenolic compound present in peppermint essential oil, has an expectorant effect, antitussive, and disinfectant activity [18]. Phenolic acids such as rosmarinic acid, cafeic acid, and flavonoids are identified in peppermint and are responsible for the high antioxidant capacity of the leaves [19,20]. Other researchers have studied the influence of microwaves on the extraction of bioactive compounds from various plants for pharmaceutical, nutraceutical, and cosmetic uses [21,22,23].

Most studies focus on the application of microwaves as a technique for extracting essential oils from peppermint [13,16,24,25]. The novelty of this study consists in the application of microwaves with the aim of making the extraction process of bioactive compounds such as polyphenols and peppermint pigments more efficient. Moreover, the change in the colour parameters of the peppermint leaves in the case of microwave treatments was highlighted and all the results were interpreted from the point of view of chemometric multivariate analysis. The main objective of these experiments was to compare the possibilities of total or partial substitution of conventional extraction technologies with the microwave extraction technology and to highlight the existing differences in the content of bioactive compounds extracted from peppermint leaves under different processing conditions. The paper thus approaches a new way of applying microwaves in the process of extracting bioactive compounds, with the aim being to develop a rapid extraction model. In addition, the study highlights the use of energy-efficient processing techniques and at the same time obtaining a high-quality final product. The environmental viability of this research depends on the quality of the peppermint.

## 2. Microwave Processing

The experimental method used is an innovative method and influences the quantity and quality of the extracted bioactive compounds. Microwave processing has the advantage of preserving the qualities of volatile oils at the same time as the generation of a microbiologically safe product [26]. Conventional technologies for processing different products are based on thermal processes that have the role of inactivating different microorganisms. Lately, the use of non-conventional methods known as green technologies is of particular interest, especially because through these methods the processing time is reduced, an increase in colour intensity as well as in the concentration of different compounds is obtained, at the same time as obtaining some quality final products with the best possible properties [27]. Microwave field processing is a technology applied in many food processes that contributes to reducing processing time, with direct influences on food preservation due to the direct antimicrobial effects on microorganisms [28]. The sensitivity of microorganisms to processing in the microwave field depends on the characteristics of the cells, such as their structure and size. The breaking of the cell membrane in the sample is accelerated at the same time with the activation and deactivation of the high-frequency electromagnetic field, that causes the appearance of a transient regime, the sample being exposed to a spectrum of frequencies, thus increasing the chances that the frequency of the electromagnetic field is identical to the breaking frequency of the cell membrane [29]. In addition, factors such as the pH of the product, the presence of water, and the electrical conductivity can have influences on some biochemical reactions that contribute to the inactivation of microorganisms [30].

For the use of high-frequency applicators, it is extremely important to know the interaction between the electromagnetic field and the material, because the electrical properties of the material to be processed are considered as components of the device’s operation [30,31]. If in the case of conventional equipment, the nature of the material to be processed has a minimal importance and contribution, in the case of high-frequency applicators, the dielectric properties of the processed materials have a major importance in the design of the device. The main characteristic of the processing of dielectric materials in microwave fields is the fact that the release of heat occurs straight inside the material to be heated.

Knowing the dielectric properties defined by the dielectric constant (*ε′*) and the loss factor (*ε*) provides a good understanding of the behaviour of different medicinal plants during processing in the microwave field:(1)$$\underline{\varepsilon }=\varepsilon^{\prime \prime }-\frac{j\sigma }{\omega \varepsilon_{0}}=\varepsilon^{\prime }-j\varepsilon^{\prime \prime }$$

The dielectric constant *ε*′ is a measure of a material’s ability to couple with high-frequency energy, while *ε* is a measure of a material’s ability to absorb heat through microwave energy. The loss factor refers to the effective loss factor, which includes the conductivity effect. These factors have a substantial influence on the dielectric properties of a material. The presence of water causes a high-frequency energy absorption, so the higher the moisture content, the more pronounced the heating effect. The decrease in the values of the loss factor *ε* is due to the interaction of dipolar and conductivity losses with the progressive decrease of free water. The decrease in dipole losses with increasing temperature is balanced by the increase in conductivity losses, which leads to constant *ε*′ values with temperature.

The power dissipated per volume unit of the material processed in the microwave field can be calculated with the relation:*P* = 2*π*⋅*f*⋅*ε*′*tanδ⋅E*^2^(2)
where: *f* is the applied frequency; *ε*′ is the absolute dielectric constant of the material; *tanδ* is the loss factor; *E* is the intensity of the electric field. 

## 3. Materials and Methods

### 3.1. Experimental Design

The experimental determinations were performed by using high-frequency field processing equipment with adjustable power between 0–900 W and a frequency of 2450 MHz. A mixed processing method was applied: classic heating of the extraction solvent, using a resistive heat source (having a maximum power of 1300 W) and an innovative one that uses microwave technologies applied straight to the sample to accelerate the extraction phenomenon of bioactive compounds, which vary in odour and flavour. The peppermint samples (*Mentha piperita* L. from a local grower in north-western Romania) subjected to the experiments were dried in the shade at room temperature and crushed using a blender, ensuring that there were no visible differences in granulation before use, so that the solvent would come into contact with as large a surface area as possible. The samples were placed in the microwave extraction applicator into a filter cartridge. The experimental plan is presented in Figure 1. The following coding for the samples were:**woMW** (untreated control sample), where the extracting process of bioactive compounds from peppermint leaves was carried out without microwaves, through infusion on 99% ethanol vapours. An extract volume of 40 mL was obtained in 40 min;**MW_t1**—continuous MW treatment was applied for 7 min. A total volume of 40 mL was obtained;**MW_t2**—the total extraction time for this case was 10 min, the time required to obtain a quantity of 40 mL of the resulting (collected) product after the extraction. During the experiment, the sample was exposed to microwave radiation, in time intervals of 30 s, and after each exposure interval, the sample was left for 60 s in the steam flow for wetting, thus creating the optimal extraction conditions. The total microwave exposure time of the sample was 210 s out of a total of 600 s of processing time. During the exposure of the sample to microwaves, an acceleration of the release of bioactive substances was obtained due to the electroporation effect that occurs following the rupture of the cell membrane. This effect was observed visually and colourimetrically with the collection of the extraction product.**MW_t3**—the process parameters mentioned above (MW_t2) were kept, but during the 10 min of the experiment, the obtained extract was collected after each exposure period of 30 s, resulting in an extraction product with a higher concentration of bioactive compounds. The total amount of extract obtained was 20 mL, respectively, 20 mL of condensed extraction solvent (residual ethanol). The extraction product obtained after processing was then subjected to biophysical and biochemical analyses.

The method proposed in this paper is an innovative one using high-frequency electromagnetic radiation (Figure 2).

During the experimental determinations, microwave energy was used for the extraction of bioactive compounds and ethanol (96.4%) was used as the extraction solvent. In order to avoid damage to the cell membrane of the sample during the experiments, the applied microwave power density was set to a value of 6 W/g. The resistive heat source was used to bring the extraction solvent to its boiling temperature so that the alcohol vapours moisten and infuse the dried and crushed peppermint leaves sample. The favourable extraction conditions for the bioactive compounds from the sample were met only after the product sample was exposed for a sufficient time to the vapours of the extraction solvent.

### 3.2. Determination of Bioactive Compounds Content and Antioxidant Capacity

#### 3.2.1. Total Phenols (TPh) Content (Folin–Ciocâlteu Assay)

Total phenolic content was determined using the Folin–Ciocalteu method [32]. In short, each peppermint extract (100 μL) was mixed with 1700 μL of distilled water, 200 μL of Folin–Ciocalteu reagent (freshly prepared, dilution 1:10, *v*/*v*), and 1000 μL of 7.5% Na_2_CO_3_ solution, and the mixture was kept at room temperature, in the dark, for 2 h. The absorbance was measured at 765 nm using a spectrophotometer (Shimadzu 1240 mini UV–Vis, Kyoto, Japan). Different concentrations of gallic acid (between 0.1–0.5 mg/mL) were used as a standard for the reference curve (y = 27.637x + 0.0069, R^2^ = 0.999) and the results were expressed in mg gallic acid equivalents (GAE)/100 g dw.

#### 3.2.2. Total Flavonoid (TFLAV) Content

The aluminium chloride colorimetric method was used for the determination of the total flavonoid content of the peppermint extracts [33]. For total flavonoid determination, quercetin was used to create the standard calibration curve. Different dilutions with ethanol of stock quercetin solution were prepared (0.1–0.5 mg/mL) and the equation regression was y = 0.8475x + 0.0065, R^2^ = 0.987. The results were expressed in mg quercetin equivalents (QE)/100 g dw.

#### 3.2.3. FRAP (Ferric Reducing Antioxidant Power) Assay

The antioxidant capacity of peppermint extract was evaluated based on the reduction of Fe^3+^ from the tripyridyltriazine Fe(TPTZ)^3+^ complex, to the blue coloured complex-Fe(TPTZ)^2+^ in an acidic medium [34]. The stock solutions consist of 300 mM acetate buffer, pH 3.6, 20 mM FeCl_3_ 6 H_2_O, and 10 mM TPTZ (2,4,6-tri-(2-pyridyl)-1,3,5-triazine) in 40 mM HCl. The working FRAP solution was freshly prepared by mixing acetate buffer, FeCl_3_ 6 H_2_O solution, and TPTZ solution in the ratio 10:1:1 (*v*/*v*/*v*). Peppermint extract (100 μL) was allowed to react with 500 μL FRAP working solution and 2 mL distilled water, for 1 h, in the dark [35]. The final product (ferrous tripyridyltriazine complex) was quantified by VIS absorption (Shimadzu 1240 mini UV–Vis) at 595 nm. The results were expressed in mmolTrolox equivalents (TE)/100 g dw [36].

#### 3.2.4. DPPH (2,2-Diphenyl-1-picryl-hydrazyl-hydrate) Assay

The radical scavenging capacity of peppermint fractions using the stable DPPH radical was determined according to the method of Brand-Williams et al., (1995) [37], with some modifications. In short, a volume of 100 μL of each peppermint fraction was mixed with 2800 μL freshly prepared methanol DPPH solution (80 μM). After exactly 30 min incubation at room temperature in darkness, the absorbance of sample was measured at 517 nm using a Shimadzu mini UV–Vis spectrophotometer. The DPPH radical scavenging activity of each fraction was expressed in mmol TE/100 g dw [38].

### 3.3. Determination of Assimilatory Pigments

The content of green pigments (Chlorophyll-a and Chlorophyll-b) and carotenoids of each ethanol fraction obtained after treatment with MW (woMW, MW_t1, MW_t2, and MW_t3) were determined according to the equations [38,39,40] presented in Table 1.

### 3.4. Statistical Analysis

The statistical analysis was to perform the univariate statistic test, one-way ANOVA (*p* = 0.05), for all studied parameters. Furthermore, multiple pairwise comparisons were done by using the Duncan post-hoc test (*p* = 0.05). The univariate tests were performed with XLStat v2014 (XLStat Inc., 244 Fifth Avenue, Suite E100, New York, NY, USA).

## 4. Result and Discussions

### 4.1. Biochemical Compounds Content—Univariate Statistical Results

Table 2 shows the phytochemical compound (TPh and FLav) content and antioxidant capacity (DPPH and FRAP) of peppermint samples. In Table 3, the content in assimilatory pigments of peppermint samples is presented. Multiple comparison tests between the analysed samples reveal that the woMW and MW_t1 do not have a statistical difference in TPh content. The samples MW_t2 and MW_t3 have no statistical difference in TPh content, and supplementary, their values are with one order of magnitude higher than the woMW samples’ one (Figure 3a). The same behaviour is achieved by the FRAP and DPPH antioxidant capacities (Figure 4). Furthermore, the same behaviour is present for the TFlav (Figure 3b), chlorophyll a, chlorophyll b, total chlorophyll, and total carotenoids content (Figure 5a–d).

The highest level of total phenols and flavonoids were obtained in the sample MW_t2 compared with the control and other samples (MW_t1 and MW_t3) (Table 2, Figure 3a). All the previous results can prescribe that the MW-treated samples perform with high levels of all analysed parameters and by this treatment for different time intervals, there are extracted different levels of active bio-compounds, chlorophyll, total carotenoids, and antioxidant capacity, but all are high above the untreated sample. Alternatively, Petkova et al. [41] investigated the microwave-assisted extraction of phenolics from peppermint and obtained 37.7 ± 0.5 mg GAE/g while claiming that microwave processing provided slight advantages in total phenols yield when compared with conventional extraction.

The total phenolic content of the soluble fraction of Medina mint was 109.98 mg GAE/g dw, while in Hasawi mint, it was 36.80 mg GAE/g dw. [19]. In another study, depending on the applied microwave conditions, the content of total phenols in *M. piperita* L. varied between 6.3703–11.3087 g GAE/100 g [42]. Differences in the total value of phenolics from the same plant species, reported in the literature may be due to different geographical origins, agro-climatic variations (climatic, seasonal, and geographical), extraction procedures, and physiological conditions of the plants [19].

The antioxidant activity of peppermint extracts obtained by different microwave treatments using ethanol as a polar solvent is mainly due to flavonoids and phenolic acids, and this was evaluated in our study by the DPPH and FRAP methods.

The range of antioxidant capacity of peppermint extracts obtained by different microwave treatments was from 0.018 ± 0.004 to 13.965 ± 2.738 mmol TE/100 g using the FRAP assay. The radical scavenging capacity of peppermint extracts towards DPPH radicals was in the range from 96.363 ± 52.628 to 4737.749 ± 1674.429 mM TE/100 g. The extract with the most potent antioxidant capacity was obtained for the MW_t2 sample. In the same sample (Table 2), the high values of TPh and TFlav were obtained, which suggests a correlation between the antioxidant capacity and polyphenol content. 

Microwave-assisted extraction was successfully applied to *Mentha piperita* L. in order to obtain a liquid extract with high contents of polyphenols, terpenoids, and antioxidant capacity [30]. Different parameters were considered such as ethanol concentration, extraction time, and liquid–solid ratio, and they demonstrate that the ethanol concentration was the most important factor for microwave extraction [30].

Following the application of microwaves, the pigments present in the peppermint were much better extracted compared to the untreated sample (woMW).

### 4.2. Colour Analysis

The one-way ANOVA for chromatic parameters emphasizes that all four samples are statistically different fromeach other (Table 4). The untreated sample performs the lightest colour (highest L* value), followed closely by MW_t1 sample, and at a huge difference with low L* values are situated the MW_t2 and MW_t3 samples. The sample hues, provided by (a*, b*) parameters, are yellow for woMW sample, green for MW_t1, and dark green for MW_t2 and MW_t3. 

The results of the correlations between all analysed parameters are presented in Table 5. The statistically significant correlations (*p* < 0.05, marked in bold face) also have a R square higher than 0.60, that consist in the average correlation level. In consequence, there are good correlations between themselves for the chromatic parameters (L*, a*, b*), active bio-compounds, antioxidant capacity, chlorophyll types, and total carotenoids. All the correlations between antioxidant capacity and L* and a* chromatic components are, as expected, statistically significant (except DPPH with L*) and negative values. The lower the L* values, the more intense the colours, and furthermore, as it results from Table 5, the antioxidant capacity is higher. The same results and behaviours are prescribed by all the chlorophyll types and total carotenoids, for the L* and a* chromatic components.

Colour differences calculated pairwise between all analysed samples (Table 6) reveal the presence of significant differences (validated by the ΔE > 5 units) in all non-identical cases. The colour analysis results are an overall biochemical compounds effect, so at this point one can confirm that all the samples are chromatically different. The a* chromatic describes the chromatic variation from the colour green (negative a* values) to the colour red (positive a* values). All the MAE samples performed negative values for the a* chromatic parameter; hence the corresponding sample colours are intense green, in contrast with the control sample that performs a positive a* chromatic value and, as a consequence, displays a yellow-green colour (Table 6). Furthermore, the antioxidant capacity, chlorophyll, and total carotenoids have positive average to strong correlations. These results are expected for the carotenoids, due to the known antioxidant capacity of this compound. On the other hand, the chlorophylls a and b usually behave as oxidants. In the analysed extract, due to a positive medium to strong correlation with the antioxidant capacities (Table 4), one can state that chlorophyll a and b have antioxidant behaviours.

Chlorophylls are cyclic tetrapyrroles and their structure and configuration influence their antioxidant activity. In one study 148, chlorophyll a was more effective of a radical quencher than chlorophyll b [43,44]; however, on the other hand, other studies have shown that chlorophyll b exhibited higher antioxidant activity compared with chlorophyll a [45]. Carotenoids, lipophilic antioxidants, are divided into two groups: carotenes (consisting only of carbon and hydrogen) and xanthophylls (carotene oxygenated products). These compounds act as antioxidants by different mechanisms or pro-oxidants depending on the environment [46]. The proposed mechanisms are: (i) the electron transfer from the conjugated polyenic chain to chlorophylls; (ii) physical quenching of singlet oxygen; (iii) antioxidant activity against peroxyl radicals through electron transfer, hydrogen abstraction, and radical addition to the ring or to the chain [46].

### 4.3. Chemometric Multivariate Analysis

The DOE involves four extracted samples: an untreated sample and three MW-treated ones, but all the samples have been extracted sequentially from the same 20 g of dried peppermint leaves. The motivation of this DOE was to emphasize the MW effect upon active bio-compounds extraction, more specifically, if some bio-compounds have MW extraction sensibility and how the molecular lattice responds to the MW treatment with time. In order to emphasize these DOE aims, a multivariate sequence was performed: principal component analysis (PCA) and multivariate analysis of variance, MANOVA (*p* = 0.05). In this way, in the end, one will be able to prescribe the sample clusters and thus which one maximises some of the analysed parameters.

Before any other discussions, it is necessary to give interpretation to the MANOVA results from Table 7. As can be noticed all statistical significances, for the multiple pairwise comparisons, they have values less than *p* = 0.05, and thus prescribe that all samples consist in singleton clusters.

Table 8 shows the summary results of the PCA; the first two principal components have above unity eigenvalues and present the cumulative explained variance of 96.63%. Despite this fact, that confirms that only PC1 and PC2 are sufficient to perform accurate interpretation of the PCA results, in the following analysis all three principal components will be considered.

The first PCA discussion is about variables grouping. The TPH, TFLAV, and FRAP consist in one group that positively correlates with the PC1 but have negative correlations with PC2. The L* and a* parameters consist in two singleton groups that negatively correlate with the PC1. The DPPH and b* consist in two singleton groups that positively correlate with the PC1 and PC2.

The second PCA discussion describes the relation between the samples and variables. The woMW and MW_t1 samples perform less oil extraction volume (low levels of b*) and have poorer antioxidant capacity than the MW_t2 and MW_t3 samples. The best sample of ethanol peppermint extract is MW_t2 that has the highest proportional antioxidant capacity, active bio compounds, chlorophyll, and the highest b* values; the high b* values correlate with high oil extraction. The MW_t3 follows MW_t2, with the highest chlorophyll levels, but with less oil and antioxidant capacity levels. After these two samples, the MW_t1 follows with the highest oil content but much lower proportional antioxidant capacity, active bio compounds, and chlorophyll levels. Then comes the woMW sample with the lowest levels of all analysed parameters. All these statements are from relative comparisons from the presented DOE.

The bioactive compounds that are responsible for the high levels of DPPH capacity are extracted earlier than others (see Figure 6, Figure 7, Figure 8 and Figure 9, timeline from woMW to MW_t3). The next released bioactive compounds are total carotenoids and chlorophyll a, and the latest are chlorophyll b and the bioactive compounds responsible for the high levels of TPh, TFlav, and FRAP. Figure 9 (i.e., 3D biplot) emphasizes the time evolution of the peppermint samples. Clearly the control sample, woMW, is in the area with the highest values of L* and a*, and the lowest values of all the other variables. Then, as time increases, the MAE peppermint samples gain high values for the other variables, until the last sample, MW_t3, performs the highest values for the TPh, TFlav, FRAP, chlorophyll a and b, total, and total carotenoids, but not for DPPH. The highest value for DPPH is present for the MW_t2 sample. These results reveal the biochemical compounds extraction behaviour of peppermint in a MAE process. 

## 5. Conclusions

During the experimental determinations, the same microwave power density applied to the samples (6 W/g) was considered. The results obtained were compared to the control sample. The extraction method with microwave input is versatile, being able to be applied both at the laboratory level and at the industrial scale, by adapting the equipment and the correlation between the microwave power density and the amount of plant material processed. The processing time is substantially reduced, thus increasing the productivity of the process and obtaining a superior quality of the extract, resulting in high values of antioxidant capacity and of the bioactive compounds, chlorophyll, and carotenoids. Thus, we can say that microwave extraction technology can successfully replace the conventional extraction technology used today.

A MAE peppermint was produced in a DOE with four successive samples. The high-frequency electromagnetic field was applied to the peppermint sample after a time interval (40 min), during which the sample was sufficiently wet. Thus, when the microwaves were applied, the sample absorbs a maximum amount of energy that contributes to the increase of bioactive compounds obtained.

All the MAE samples performed higher levels of antioxidant capacities (TPh, TFlav, FRAP, and DPPH), chlorophyll a, b, total, and total carotenoid content than the untreated (control) sample. Furthermore, in this way, the peppermint extract is a functional product with bio-medical applications. 

The multivariate statistical results explain how the biochemical compounds are extracted in the peppermint MAE process. According to the timeline from woMW to MW_t3, the bioactive compounds that are responsible for the high level of DPPH capacity are extracted first, then the total carotenoids and chlorophyll a are released, and lastly the chlorophyll b and the bioactive compounds responsible for the high levels of TPh, TFlav, and FRAP are released. Furthermore, all the presented results prove that the successive MAE process creates peppermint samples with high levels of bioactive compounds, but with low energy consumption due to its mainly successive strategy that cancels the intermediary steps.

We can say that by using the energy of the high-frequency field, a better electroporation of the cell membrane of the peppermint sample is obtained, which determines the release of bioactive compounds and assimilatory pigments. The level of extracted compounds obtained is higher in a much shorter processing time (30 s).

Further research will address the influence of the microwave power density applied to the sample. There is thus the possibility to study the influence of this parameter on the extraction process and the quality of the obtained product. A relationship can thus be established between the power density applied to the sample and the amount of bioactive compounds released in the experiment.

## Figures and Tables

**Figure 1 materials-15-07789-f001:**
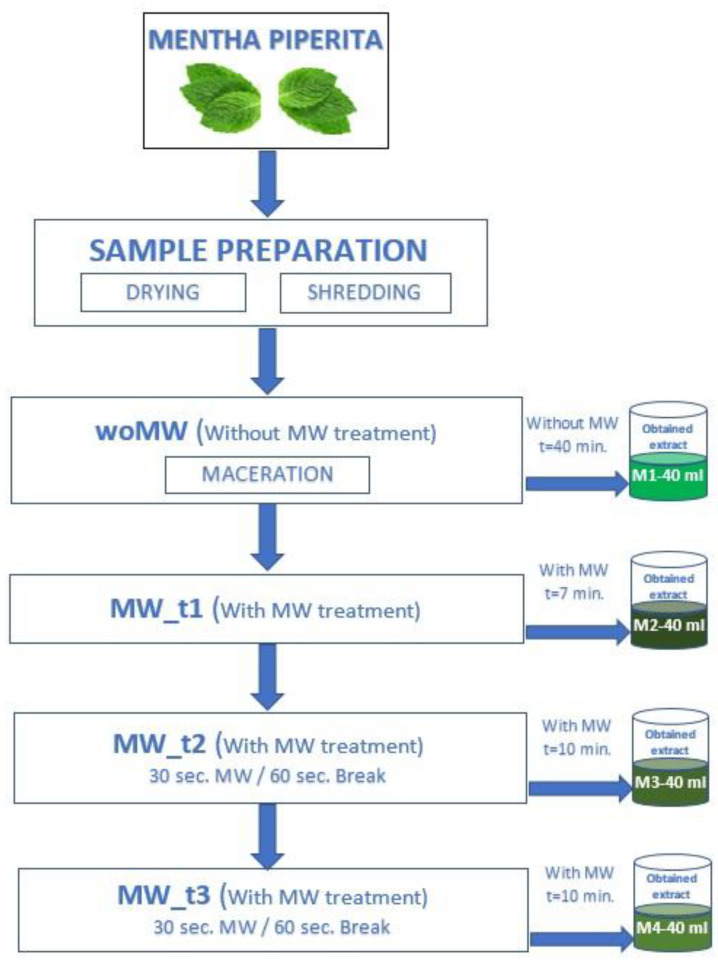
Experimental design (DOE).

**Figure 2 materials-15-07789-f002:**
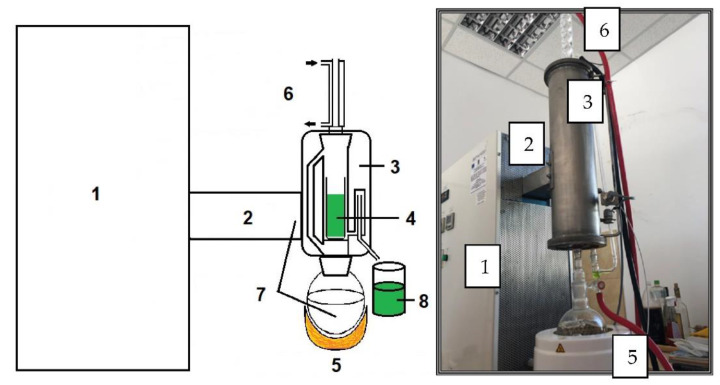
The experimental equipment for the extraction of bioactive compounds from peppermint. 1—microwave generator; 2—microwave guide; 3—applicator; 4—peppermint sample; 5—extraction solvent heating nest; 6—steam condensers; 7—extraction solvent; 8—extract.

**Figure 3 materials-15-07789-f003:**
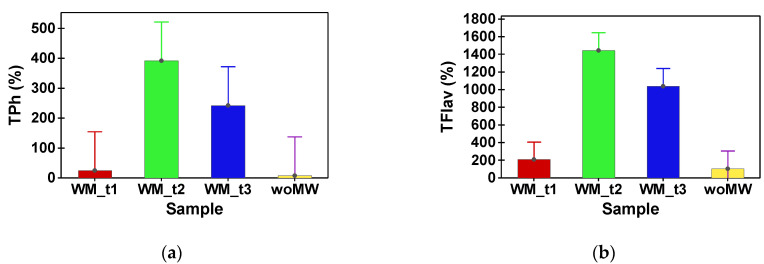
Ratios (%) between peppermint samples (MW_t1, MW_t2, and MW_t3) and the untreated sample, woMW. (**a**) Total polyphenols (TPh). (**b**) Total flavonoids (TFlav).

**Figure 4 materials-15-07789-f004:**
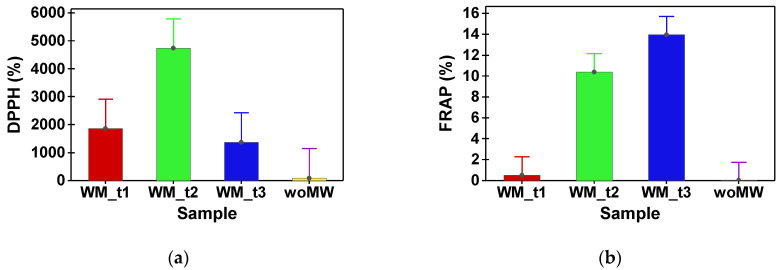
Ratios (%) between peppermint samples (MW_t1, MW_t2, and MW_t3) and the untreated sample, woMW. (**a**) Antioxidant capacity using DPPH assay. (**b**) Antioxidant capacity using FRAP assay.

**Figure 5 materials-15-07789-f005:**
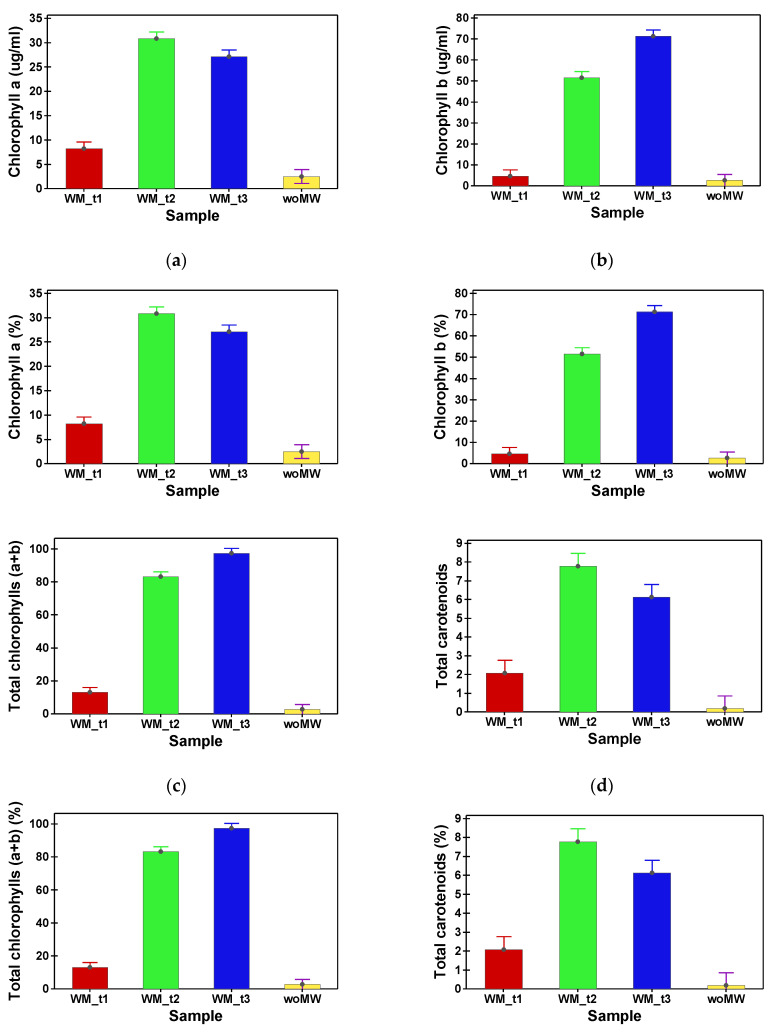
Ratios (%) between MW samples (MW_t1, MW_t2, and MW_t3) and the untreated sample, woMW. Chlorophyll a (**a**), chlorophyll b (**b**), total chlorophyll (**c**), and total carotenoids (**d**).

**Figure 6 materials-15-07789-f006:**
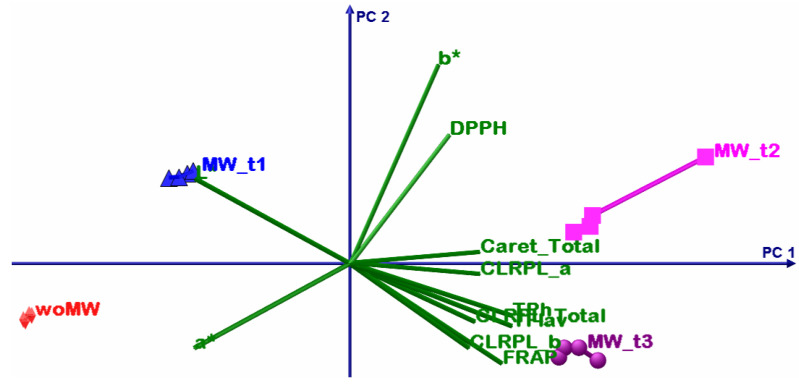
PCA biplot with PC1 (81.04%) and PC2 (15.59%) principal axes.

**Figure 7 materials-15-07789-f007:**
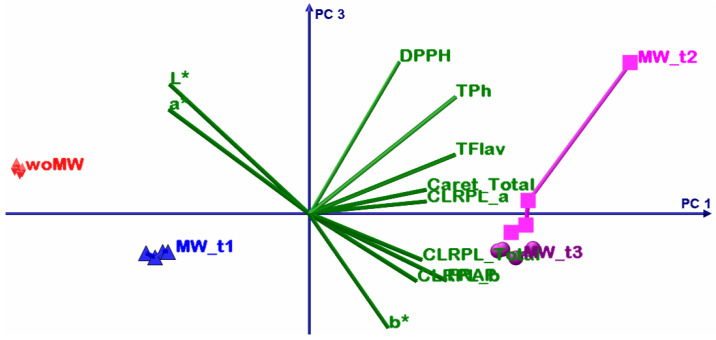
PCA biplot with PC1 (81.04%) and PC3 (3.35%) principal axes.

**Figure 8 materials-15-07789-f008:**
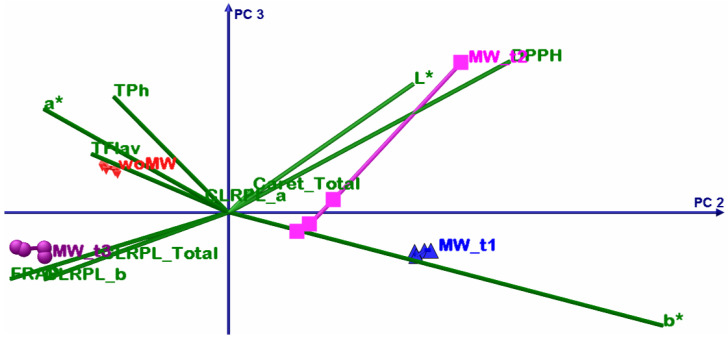
PCA biplot with PC2 (15.59%) and PC3 (3.35%) principal axes.

**Figure 9 materials-15-07789-f009:**
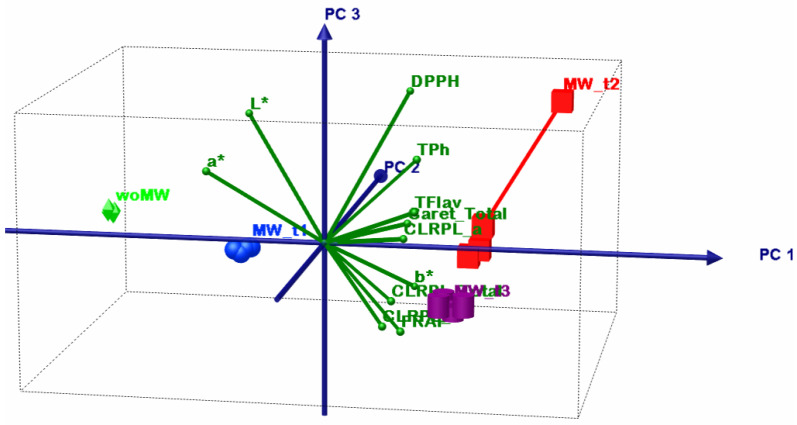
PCA biplot with PC1 (81.04), PC2 (15.59%), and PC3 (3.35%) principal axes.

**Table 1 materials-15-07789-t001:** Equations to determine concentrations (μg/mL) of chlorophyll a, chlorophyll b, and total carotenoids from ethanol extract of peppermint.

Solvent	Equations
Ethanol	Chlorophyll a = 13.36 A664 − 5.19 A649
	Chlorophyll b = 27.43 A649 − 8.12 A664
	Total carotenoids = (1000 A470 − 2.13 Ca- − 97.63 Cb)/209

**Table 2 materials-15-07789-t002:** Mean results of total polyphenols (TPh), total flavonoid (TFlav) content, and antioxidant capacity (FRAP and DPPH) results from one-way ANOVA (*p* = 0.05) *.

Samples	TPh (mg GAE/100 g)	TFlav (mg QE/100 g)	FRAP (mmol TE/100 g)	DPPH (mmol TE/100 g)
woMW	8.089 b ± 2.745	106.747 c ± 7.446	0.018 c ± 0.004	96.363 c ± 52.628
MW_t1	25.000 b ± 1.992	208.515 c ± 17.648	0.515 c ± 0.075	1861.150 b ± 63.083
MW_t2	391.687 a ± 20.537	1443.513 a ± 305.905	10.409 b ± 0.526	4737.749 a ± 167.429
MW_t3	242.461 a ± 30.648	1038.737 b ± 91.743	13.965 a ± 2.738	1373.741 bc ± 70.575

* Different letters across columns for each variable denote statistically significant differences between the samples’ means—pairwise comparisons were done with the Duncan post-hoc test (*p* = 0.05). Results are expressed as mean ± standard deviation (*n* = 4).

**Table 3 materials-15-07789-t003:** Mean results of chlorophyll a, chlorophyll b, total chlorophyll, and total carotenoid content results from one-way ANOVA (*p* = 0.05) *.

Samples	Chlorophyll a (µg/mL)	Chlorophyll b (µg/mL)	Total Chlorophyll (µg/mL)	Total Carotenoids (µg/mL)
woMW	2.494 d ± 0.794	2.642 c ± 1.468	5.136 d ± 0.170	0.189 d ± 0.027
MW_t1	8.194 c ± 0.742	4.695 c ± 0.187	13.157 c ± 0.679	2.084 c ± 0.737
MW_t2	30.833 a ± 1.673	51.582 b ± 2.555	83.150 b ± 3.209	7.792 a ± 0.602
MW_t3	27.101 b ± 1.048	71.341 a ± 3.593	97.345 a ± 3.527	6.130 b ± 0.516

* Different letters across columns for each variable denote statistically significant differences between the samples’ means—pairwise comparisons were done with the Duncan post-hoc test (*p* = 0.05). Results are expressed as mean ± standard deviation (*n* = 4).

**Table 4 materials-15-07789-t004:** Mean results of chromatic parameters of RGB and CIE L*, a*, b* spaces, from one-way ANOVA (*p* = 0.05) *.

Samples	R	G	B	L*	a*	b*
**woMW**	243.00 a ± 1.000	240.75 a ± 1.299	230.50 a ± 1.500	96.44 a ± 1.712	4.31 a ± 1.711	23.44 d ± 1.562
**MW_t1**	209.75 b ± 1.479	214.25 b ± 1.090	125.25 b ± 1.479	86.00 b ± 1.572	−10.67 b ± 1.583	66.21 a ± 1.128
**MW_t2**	81.00 c ± 1.732	95.00 c ± 1.581	8.75 c ± 1.785	39.44 c ± 1.562	−18.22 c ± 0.807	62.01 b ± 1.726
**MW_t3**	49.25 d ± 0.433	60.00 d ± 1.732	4.00 d ± 1.414	24.69 d ± 1.371	−12.02 b ± 1.246	39.77 c ± 1.453

* Different letters across columns for each variable denotes statistically significant differences between the samples’ means—pairwise comparisons were done with the Duncan post-hoc test (*p* = 0.05). Results are expressed as mean ± standard deviation (*n* = 4).

**Table 5 materials-15-07789-t005:** Correlations between all parameters of all samples.

woMW	L*	a*	b*	TPh	TFlav	FRAP	DPPH	Chlorophyll a	Chlorophyll b	Total Chlorophyll	Total Carotenoids
**L***	**1.000**	**0.728**	−0.196	**−0.720**	**−0.870**	**−0.965**	−0.462	**−0.952**	**−0.987**	**−0.994**	**−0.908**
**a***	**0.728**	**1.000**	**−0.784**	**−0.636**	**−0.758**	**−0.643**	**−0.745**	**−0.821**	**−0.643**	**−0.724**	**−0.859**
**b***	−0.196	**−0.784**	**1.000**	0.319	0.336	0.104	**0.688**	0.368	0.086	0.196	0.445
**TPh**	**−0.720**	**−0.636**	0.319	**1.000**	**0.941**	**0.732**	**0.856**	**0.826**	**0.692**	**0.755**	**0.854**
**TFlav**	**−0.870**	**−0.758**	0.336	**0.941**	**1.000**	**0.836**	**0.799**	**0.951**	**0.833**	**0.893**	**0.952**
**FRAP**	**−0.965**	**−0.643**	0.104	**0.732**	**0.836**	**1.000**	0.423	**0.912**	**0.982**	**0.966**	**0.886**
**DPPH**	−0.462	**−0.745**	**0.688**	**0.856**	**0.799**	0.423	**1.000**	**0.677**	0.382	**0.500**	**0.746**
**Chlorophyll a**	**−0.952**	**−0.821**	0.368	**0.826**	**0.951**	**0.912**	**0.677**	**1.000**	**0.923**	**0.968**	**0.982**
**Chlorophyll b**	**−0.987**	**−0.643**	0.086	**0.692**	**0.833**	**0.982**	0.382	**0.923**	**1.000**	**0.987**	**0.876**
**Total** **chlorophyll**	**−0.994**	**−0.724**	0.196	**0.755**	**0.893**	**0.966**	**0.500**	**0.968**	**0.987**	**1.000**	**0.928**
**Total** **carotenoids**	**−0.908**	**−0.859**	0.445	**0.854**	**0.952**	**0.886**	**0.746**	**0.982**	**0.876**	**0.928**	**1.000**

Note: Values in bold are different from 0 with a significance level *p* = 0.05 (*n* = 16).

**Table 6 materials-15-07789-t006:** Mean results of colour differences, ΔE in CIE L*, a*, b* coordinates, between all the samples. Rendered colours are generated with RGB values.

ΔE	woMW	MW_t1	MW_t2	MW_t3	Rendered Colour
**woMW**	**0.75**	47.26	73.14	76.19	
**MW_t1**	47.26	**0.60**	46.85	67.09	
**MW_t2**	73.14	46.85	**0.55**	28.44	
**MW_t3**	76.19	67.09	28.44	**0.20**	

**Table 7 materials-15-07789-t007:** Statistically significance values from MANOVA (PCA) test with selected parameters.

*p*-Values	woMW	MW_t1	MW_t2	MW_t3
**woMW**		<0.001	<0.001	<0.001
**MW_t1**	<0.001		<0.001	<0.001
**MW_t2**	<0.001	<0.001		<0.001
**MW_t3**	<0.001	<0.001	<0.001	

**Table 8 materials-15-07789-t008:** Summary of the PCA analysis of selected parameters.

PC	Eigenvalue	Explained Variance (%)
**PC1**	8.915	81.04
**PC2**	1.715	15.59
**PC3**	0.368	3.35

## Data Availability

Not applicable.

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
