# Peer review of "The Influence of Microwave Treatments on Bioactive Compounds and Antioxidant Capacity of Mentha piperita L."

_materials, 2022, doi:10.3390/ma15217789_

Round 1

Reviewer 1 Report

This paper proposes a new method of extracting bioactive compounds (phenols, flavonoids, chlorophyll) from peppermint using high frequency electromagnetic radiation. Control variable method was used to compare the effects of microwave extraction technology and traditional extraction technology on the quality of the final bioactive compounds, and considerable experimental results were obtained. However, this paper still has the following deficiencies: 1) The introduction elaborates too much on the research background of “medicinal plants”. In this part, the main objective should be “peppermint”, focusing on its research background, value and current processing, and pointing out the existing problems of current research. In addition, the advantages of microwave extraction technology should be highlighted in comparison with conventional extraction techniques. Based on previous studies, the possibility of using this method to solve the current problem is demonstrated. You may also briefly describe why better research results can be achieved using it. 2) In this part of the experimental design, what is the difference between the treatment conditions of MW_t2 and MW_t3 and what is the significance of MW_t2 and MW_t3 as control groups? What are the factors that affect the microwave extraction technique? What are the factors that affect the extraction of bioactive compounds? How did you control these variables? A clear explanation is needed in this regard.  Fig 2 is not clear, a better one should be provided, and you can try to beautify it further.  In my opinion, the title of 2.2 should be changed from “Determination of bioactive compounds and antioxidant capacity” to “Determination of bioactive compounds content and antioxidant capacity”.  Some sentences contain grammatical mistakes or are not complete sentences, such as, in page 3, Experimental design, “the microwaves where applied in time intervals of 30 seconds with a break of 60 seconds with elimination of residual alcohol (approx. 4 ml)” and so on.  In the second part of the article-Materials and methods, the secondary headings 2.1 are bolded, the others are not, and the formatting needs to be unified. 3) In result and discussions, it is suggested that the background of microwave processing should be mentioned in the introduction, which can be briefly described in this part. Furthermore, according to your references, it can be seen that the extraction technique has been successfully applied to the extraction of peppermint bioactive compounds by previous authors, what is the outstanding contribution of this study compared with other studies in this field? 4) In the conclusion section, with the above experiments and data analysis, you should evaluate whether microwave extraction technology can partially or completely replace the conventional extraction technology or how likely it is. What are the limitations of the present study or what can be done in more depth? Conclusion needs more in it, as it's more of an afterthought. The authors are suggested to highlight important findings and include afterthought of this work.

Reviewer 2 Report

This manuscript is entitled "The Influence of Microwave Treatments on Bioactive Compounds and Antioxidant Capacity of Mentha Piperita". These data are interesting, but some points still need to correct before publication.

1.     Please check spelling mistakes and the English language throughout the text.

2.     Abstract: please rewrite the main results and the purpose

3.     Please check all figures and correct them properly. (Especially Figure 10).

4.     Introduction: The novelty and the advance added to the area must be clearly stated. Particularly Introduction could be enlarged. These things are missing.

5.     Please check all units in order to be similar

6.     Environmental viability assessment should be added.

7.     References can be added from the host journal.

8.     Conclusion: please add the key points with the further implication

Round 2

Reviewer 1 Report

Accepted